**Data Availability Statement:** All relevant data are within the paper and its Supporting Information files.

# Effect of hemodialysis on short-term outcomes after colon cancer surgery

Toshio Shiraishi[1,2], Tetsuro Tominaga[1]*, Takashi Nonaka[1], Shintaro Hashimoto[3], Kiyoaki Hamada[3], Masato Araki[3], Yorihisa Sumida[3], Hiroaki Takeshita[4], Hidetoshi Fukuoka[5], Hideo Wada[6], Kazuo To[6], Mariko Yamashita[7], Kenji Tanaka[7], Terumitsu Sawai[1], Takeshi Nagayasu[1]

1 Department of Surgical Oncology, Nagasaki University Graduate School of Biomedical Science, Nagasaki, Japan, 2 Department of Surgery, Sasebo Chuo Hospital, Sasebo, Nagasaki, Japan, 3 Department of Surgery, Sasebo City General Hospital, Sasebo, Nagasaki, Japan, 4 Department of Surgery, National Hospital Organization Nagasaki Medical Center, Nagasaki, Japan, 5 Department of Surgery, Isahaya General Hospital, Isahaya, Nagasaki, Japan, 6 Department of Surgery, Ureshino Medical Center, Ureshino, Saga, Japan, 7 Department of Surgery, Saiseikai Nagasaki Hospital, Nagasaki, Japan

☯ These authors contributed equally to this work.

* tetsuro.tominaga@nagasaki-u.ac.jp

## Abstract

### Background

Hemodialysis patients who undergo surgery have a high risk of postoperative complications. The aim of this study was to determine whether colon cancer surgery can be safely performed in hemodialysis patients.

### Methods

This multicenter retrospective study included 1372 patients who underwent elective curative resection surgery for colon cancer between April 2016 and March 2020.

### Results

Of the total patients, 19 (1.4%) underwent hemodialysis, of whom 19 (100%) had poor performance status and 18 had comorbidities (94.7%). Minimally invasive surgery was performed in 78.9% of hemodialysis patients. The postoperative complication rate was significantly higher in hemodialysis than non-hemodialysis patients (36.8% vs. 15.5%, p = 0.009). All postoperative complications in the hemodialysis patients were infectious type. Multivariate analysis revealed a significant association of hemodialysis with complications (odds ratio, 2.9362; 95%CI, 1.1384–7.5730; p = 0.026).

### Conclusion

Despite recent advances in perioperative management and minimally invasive surgery, it is necessary to be aware that short-term complications can still occur, especially infectious complications in hemodialysis patients.

**Funding:** The authors received no specific funding for this work.

**Competing interests:** The authors have declared that no competing interests exist.

## Introduction

Colorectal cancer is the third most commonly diagnosed cancer worldwide and the number of patients continues to increase; there were an estimated 1.8 million new cases in 2018 [1]. The increased use of minimally invasive surgery and advances in perioperative management have improved perioperative performance following surgery for colorectal cancer [2–5].

The prevalence of chronic kidney disease has been estimated as 9.1%, and the number of patients receiving renal replacement therapy exceeded 2.5 million worldwide in 2017 [6]. Although long-term survival has improved following advances in oral medications and medical devices, including hemodialysis (HD), HD patients encounter additional obstacles because they often have numerous other systemic comorbidities and are more likely to suffer postoperative complications [7].

In recent years, the number of patients requiring long-term HD has increased; accordingly, the number of HD patients who develop colon cancer that requires surgery has also increased [8]. However, few studies have reported surgery for colon cancer in HD patients [8–11]. The aim of this multicenter retrospective study is to determine whether colon cancer surgery can be safely performed in HD patients.

## Materials and methods

This study design was approved by the Nagasaki University Hospital Clinical Research Ethics Committee (Permission number: 16062715–2). We gave the patients written and oral explanation using the consent statement and obtained consent signature document. This multicenter, retrospective study was designed by the Nagasaki Colorectal Oncology Group (NCOG). We retrospectively reviewed the medical records of 1387 patients, including incomplete clinicopathological data, of consecutive colon cancer patients who underwent curative resection between April 2016 and March 2020 at a participating hospital (Nagasaki University Hospital, Sasebo City General Hospital, Nagasaki Medical Center, Isahaya General Hospital, Ureshino Medical Center, and Saiseikai Nagasaki Hospital). After excluding patients with synchronous colon cancer (n = 15), 1372 patients were eligible for analysis. The study protocol was reviewed and approved by the Clinical Research Review Boards of all participating hospitals.

The patients were divided into three groups according to renal function, as follows: HD group (n = 19), renal dysfunction group (n = 59), and normal group (n = 1294). The renal dysfunction and normal patients together were classified as the non-HD group. In Japan, chronic kidney disease is generally defined as persistence for ≥3 months of one or more of the following: obvious kidney disease in pathological examination, estimated glomerular filtration rate <60 ml/min/1.73m$^2$ in blood test, or proteinuria in urinalysis. In this study, renal dysfunction was defined as chronic kidney disease without HD.

We compared the clinical features among the groups and collected the following data: sex, age at surgery, body mass index (BMI), American Society of Anesthesiologists (ASA)-performance status (PS), comorbidities, tumor location, tumor size, preoperative chemotherapy, preoperative stent placement, clinical T status, clinical N status, and distant metastasis. Comorbidities included such as hypertension, diabetes, heart disease, and cerebrovascular disease. Regarding tumor location, we defined right-side colon as from the cecum to the transverse colon, and left-side colon as from the descending to the sigmoid colon. The following surgical and pathological data were collected: approach, combined resection of adjacent organs, number of retrieved lymph nodes, tumor size, operation time, estimated blood loss, postoperative complications, and postoperative hospital stay. Table 1 lists all comorbidities that were assessed, including hypertension, diabetes, dyslipidemia, heart disease, and cerebrovascular disease. Table 2 shows the presence or absence of the same comorbidities as in

**Table 1. Patient characteristics.**

|  | All patients (*n* = 1372) (%) |
|---|---|
| Sex |  |
| Male | 714 (52.0) |
| Female | 658 (48.0) |
| Age, y (range) | 72 (24–96) |
| Body mass index, kg/m$^2$ (range) | 22.1 (12.9–42.0) |
| ASA performance status |  |
| 1 | 429 (31.3) |
| 2 | 815 (59.4) |
| 3– | 118 (8.6) |
| Comorbidity, present | 874 (63.7) |
| Renal function |  |
| HD | 19 (1.4) |
| Renal dysfunction | 59 (4.3) |
| Normal | 1294 (94.3) |
| Tumor location |  |
| Right-side colon | 715 (52.1) |
| Left-side colon | 656 (47.8) |
| Tumor size, mm (range) | 40 (0.6–190) |
| Preoperative chemotherapy | 43 (3.1) |
| Preoperative stent placement | 119 (8.7) |
| Clinical T status |  |
| 1 | 312 (22.7) |
| 2 | 187 (13.6) |
| 3 | 577 (42.1) |
| 4 | 278 (20.3) |
| Clinical N status |  |
| 0 | 771 (56.2) |
| 1 | 352 (25.7) |
| 2 | 190 (13.8) |
| 3 | 40 (2.9) |
| Distant metastasis, present | 173 (12.6) |
| Approach |  |
| Open | 197 (14.4) |
| Laparo | 1173 (85.5) |
| Combined resection |  |
| None | 1267 (92.3) |
| Yes | 99 (7.2) |
| Retrieved lymph nodes, n (range) | 16 (0–115) |
| Operation time, min (range) | 211 (55–725) |
| Blood loss, mL (range) | 25 (0–3935) |
| Postoperative complications, CD $\geq$ 2 | 217 (15.8) |
| Hospital stay, days (range) | 13 (3–155) |

Data are presented as the number of patients or the median (range).

*ASA*, American Society of Anesthesiologists; *HD*, hemodialysis; *CD*, Clavien–Dindo.

**Table 2. Comparison of clinicopathological characteristics by renal function group.**

| | HD (*n* = 19) (1.4%) | Renal dysfunction (*n* = 59) (4.3%) | Normal (*n* = 1294) (94.3%) | p-value | HD vs Non-HD p-value |
|---|---|---|---|---|---|
| Sex | | | | 0.518 | 0.959 |
| Male | 10 (52.6) | 35 (59.3) | 669 (51.7) | | |
| Female | 9 (47.4) | 24 (40.7) | 625 (48.3) | | |
| Age, y (range) | 77 (60–86) | 81 (58–96) | 71 (24–96) | <0.001 | 0.282 |
| Body mass index, kg/m$^2$ (range) | 21.7 (16.2–31.8) | 22.5 (16.0–37.6) | 22.0 (12.9–42.0) | 0.584 | 0.482 |
| ASA performance status | | | | <0.001 | <0.001 |
| 1 | 0 (0) | 11 (18.6) | 418 (32.3) | | |
| 2 | 0 (0) | 33 (55.9) | 782 (60.4) | | |
| 3– | 19 (100) | 15 (25.5) | 84 (6.5) | | |
| Comorbidity | | | | <0.001 | 0.012 |
| None | 1 (5.3) | 7 (11.9) | 400 (30.9) | | |
| Yes | 18 (94.7) | 52 (88.1) | 804 (62.1) | | |
| Tumor location | | | | 0.894 | 0.966 |
| Right-side colon | 10 (52.6) | 29 (49.2) | 676 (52.2) | | |
| Left-side colon | 9 (47.4) | 30 (50.8) | 617 (47.7) | | |
| Preoperative chemotherapy | | | | 0.508 | 0.429 |
| None | 19 (100) | 56 (94.9) | 1250 (96.6) | | |
| Yes | 0 (0) | 3 (5.1) | 40 (3.1) | | |
| Preoperative stent placement | | | | 0.399 | 0.176 |
| None | 19 (100) | 54 (91.5) | 1180 (91.2) | | |
| Yes | 0 (0) | 5 (8.5) | 114 (8.8) | | |
| Clinical T status | | | | 0.164 | 0.368 |
| 1 | 4 (21.1) | 16 (27.1) | 292 (22.6) | | |
| 2 | 5 (26.3) | 3 (5.1) | 179 (13.8) | | |
| 3 | 8 (42.1) | 31 (52.5) | 538 (41.6) | | |
| 4 | 2 (10.5) | 9 (15.3) | 267 (20.6) | | |
| Clinical N status | | | | 0.300 | 0.267 |
| 0 | 13 (68.4) | 38 (64.4) | 720 (55.6) | | |
| 1 | 6 (31.6) | 12 (20.3) | 334 (25.8) | | |
| 2 | 0 (0) | 9 (15.3) | 181 (14.0) | | |
| 3 | 0 (0) | 0 (0) | 40 (3.1) | | |
| Distant metastasis | | | | 0.085 | 0.093 |
| Absence | 19 (100) | 55 (93.2) | 1108 (85.6) | | |
| Presence | 0 (0) | 4 (6.8) | 169 (13.1) | | |
| Approach | | | | 0.591 | 0.404 |
| Open | 4 (21.1) | 10 (16.9) | 183 (14.1) | | |
| Laparo | 15 (78.9) | 49 (83.1) | 1109 (85.7) | | |
| Combined resection | | | | 0.372 | 0.220 |
| None | 19 (100) | 56 (94.9) | 1192 (92.1) | | |
| Yes | 0 (0) | 3 (5.1) | 96 (7.4) | | |
| Retrieved lymph nodes, n (range) | 18 (3–76) | 14 (0–45) | 16 (0–115) | 0.015 | 0.933 |
| Tumor size, mm (range) | 30 (8–80) | 42 (4.5–90) | 40 (0.6–190) | 0.920 | 0.482 |
| Operation time, min (range) | 237 (137–366) | 198 (84–461) | 211 (55–725) | 0.047 | 0.039 |
| Open | 176 (137–366) | 138 (84–221) | 207 (55–719) | 0.014 | 0.303 |
| Laparo | 250 (160–327) | 197 (84–461) | 211 (82–725) | 0.009 | 0.005 |
| Blood loss, mL (range) | 30 (0–243) | 30 (0–566) | 25 (0–3935) | 0.579 | 0.528 |
| Postoperative complications, | | | | 0.070 | 0.009 |

*(Continued)*

**Table 2.** (Continued)

| | HD (*n* = 19) (1.4%) | Renal dysfunction (*n* = 59) (4.3%) | Normal (*n* = 1294) (94.3%) | p-value | HD vs Non-HD p-value |
|---|---|---|---|---|---|
| CD <2 | 12 (63.2) | 48 (81.4) | 1078 (83.3) | | |
| CD ≥2 | 7 (36.8) | 11 (18.6) | 199 (15.4) | | |
| Hospital stay, days (range) | 17 (9–155) | 15 (8–89) | 13 (3–125) | <0.001 | 0.005 |

Data are presented as the number of patients or the median (range).

*HD*, hemodialysis; *ASA*, American Society of Anesthesiologists; *CD*, Clavien–Dindo.

Differences in categorical variables were compared using Fisher's exact test or the chi-squared test, as appropriate. Differences in continuous variables were analyzed with One-way ANOVA (analysis of variance).

Table 1, except for HD and renal dysfunction. Postoperative complications were defined as those occurring within 30 days of the primary surgery.

Anastomotic leak was defined as stool-like changes in abdominal drainage or abscess formation around the anastomotic site, confirmed by imaging examination. Pneumonia was defined as the presence of respiratory symptoms and imaging findings of pneumonia. Surgical site infection (SSI) was defined as that in an incision, organ, or abdominal cavity affected by surgery. Urinary tract infection was defined as pyuria according to urinalysis. Among the postoperative complications in the HD group, urinary tract infection was defined as patients with urine output who had findings of urinary tract infection. Pseudomembranous colitis and MRSA colitis in which the cause could be identified was termed colitis. Paralytic ileus was defined as the presence of abdominal distension or vomiting, with imaging findings of a dilated intestinal tract suggestive of ileus. Anastomotic bleeding was defined as the presence of melena after surgery and that was confirmed by colonoscopy. Delirium was defined as a transient psychiatric symptom that developed postoperatively. Thrombosis was defined as symptoms of lower limb pain and dyspnea, elevated D-dimer in blood tests, with thrombus identified by ultrasonography or contrast CT, including deep vein thrombosis and pulmonary embolism. Lymphorrhea was defined as milky white ascites with high triglyceride levels and without infectious from abdominal drainage. "Other" included postoperative cholecystitis, neuroleptic malignant syndrome, or stomach volvulus. CD > 2 complications were defined as those requiring antibiotic treatment, surgery or advanced medical care; and death. This time, we focused on CD grades 2 and higher because these are clinically problematic.

Statistical analysis was performed using Bell Curve for Excel software, version 3.00. The data are presented as median values with ranges. Differences in categorical variables were compared using Fisher's exact test or the chi-squared test. Differences in continuous variables were analyzed with One-way ANOVA (analysis of variance).

Multivariate analysis using binary logistic regression analysis was used to identify the independent risk factors for postoperative complications. Clinical variables with a p value <0.2 in the univariate analysis were included in the multivariate analysis. All p values <0.05 were considered significant.

## Results

Table 1 lists the clinicopathological characteristics of the 1372 patients. The study population included 714 male and 658 female patients, with median age of 72 (range, 24–96) years. The median BMI was 22.1 (range, 12.9–42.0) kg/m$^2$, 933 patients (68.0%) had poor PS (PS≥2), 874 patients (63.7%) had preoperative comorbidities, and 1173 patients (85.5%) received laparoscopic surgery. The median operation time and blood loss were 211 (range, 55–725) min and

**Table 3. Postoperative complications (CD ≥2).**

|  | HD (*n* = 19) (%) | Renal dysfunction (*n* = 59) (%) | Normal (*n* = 1294) (%) |
|---|---|---|---|
| Postoperative complication, CD ≥ 2 | 7 (36.8) | 11 (18.6) | 218 (16.8) |
| Infectious complications | 7 (36.8) | 11 (18.6) | 131 (10.1) |
| Anastomotic leak | 3 (15.8) | 5 (8.5) | 48 (3.7) |
| Pneumonia | 2 (10.5) | - | 13 (1.0) |
| SSI | 1 (5.3) | 4 (6.8) | 50 (3.9) |
| Urinary tract infection | 1 (5.3) | 1 (1.7) | 8 (0.6) |
| Intraperitoneal abscess | - | 1 (1.7) | 4 (0.3) |
| Pseudomembranous colitis | - | - | 7 (0.5) |
| MRSA colitis | - | - | 1 (0.1) |
| Non-infectious complications | - | - | 87 (6.7) |
| Paralytic ileus | - | - | 45 (3.5) |
| Anastomotic bleeding | - | - | 9 (0.7) |
| Delirium | - | - | 4 (0.3) |
| Lymphorrhea | - | - | 4 (0.3) |
| Thrombosis | - | - | 3 (0.2) |
| Other | - | - | 22 (1.7) |

Differences in categorical variables were compared using Fisher's exact test or the chi-squared test, as appropriate.

*HD*, hemodialysis; *CD*, Clavien–Dindo; *SSI*, surgical site infection.

25 (range, 0–3935) mL, respectively. Postoperative complications (CD ≥2) occurred in 217 patients (15.8%). The median postoperative hospital stay was 13 (range 3–155) days.

Table 2 shows the comparison of clinicopathological differences among the three renal function groups. Age, ASA-PS, preoperative comorbidities excluding renal dysfunction, number of retrieved lymph nodes, operation time, postoperative complications, and postoperative hospital stay showed significant difference among the groups. There were also significant differences in postoperative complications (CD ≥2) between the HD and non-HD groups (p = 0.009). The rate of patients with postoperative complications (CD >2) was 36.8% (n = 7) in the HD group, 18.6% (n = 11) in the renal dysfunction group, and 15.4% (n = 199) in the normal group. There were no significant differences in any other factors among the three groups. There were significant differences between the HD and non-HD groups in terms of ASA-PS, preoperative comorbidities excluding renal dysfunction, operation time, postoperative complications, and postoperative hospital stay.

Table 3 lists the postoperative complications (CD ≥2) in the three groups. The overall postoperative complication (CD ≥2) rate was 17.2% (n = 236): 7 (36.8%) in the HD group (n = 19), 11 (18.6%) in the renal dysfunction group (n = 59) and 218 (16.8%) in the normal group (n = 1294). All postoperative complications in the HD group (n = 7) were infectious complications: anastomotic leak (n = 3), pneumonia (n = 2), surgical site infection (SSI) (n = 1), and urinary tract infection (n = 1). All those in the renal dysfunction group (n = 11) were infectious complications, including anastomotic leak (n = 5), SSI (n = 4), urinary tract infection (n = 1), and intraperitoneal abscess (n = 1). Those in the normal group were SSI (n = 50), anastomotic leak (n = 48), and paralytic ileus (n = 45).

Table 4 lists the results of univariate and multivariate analyses. Multivariate analysis revealed HD (odds ratio, 2.9362; 95%CI, 1.1384–7.5730; p = 0.026) as a risk factor significantly associated with complications.

**Table 4. Clinicopathological factors predicting postoperative complications in colon cancer patients.**

| | Univariate analysis | | | Multivariate analysis | | |
|---|---|---|---|---|---|---|
| | **Odds ratio** | **95%CI** | **p value** | **Odds ratio** | **95%CI** | **p value** |
| Sex | | | 0.908 | | | |
| Female | 1 | | | | | |
| Male | 1.0172 | 0.7606–1.3604 | | | | |
| Age, y | | | 0.426 | | | |
| <70 | 1 | | | | | |
| ≥70 | 1.1278 | 0.8385–1.5169 | | | | |
| Body mass index, kg/m$^2$ | | | 0.242 | | | |
| <25 | 1 | | | | | |
| ≥25 | 0.8030 | 0.5561–1.1595 | | | | |
| ASA performance status | | | 0.610 | | | |
| 1 | 1 | | | | | |
| 2– | 1.0858 | 0.7913–1.4899 | | | | |
| HD | | | 0.018 | | | 0.026 |
| None | 1 | | | 1 | | |
| Yes | 3.1278 | 1.2173–8.0369 | | 2.9362 | 1.1384–7.5730 | |
| Tumor location | | | 0.893 | | | |
| Right-side colon | 1 | | | | | |
| Left-side colon | 1.0201 | 0.7629–1.3640 | | | | |
| Preoperative chemotherapy | | | 0.945 | | | |
| None | 1 | | | | | |
| Yes | 1.0292 | 0.4519–2.3441 | | | | |
| Clinical T status | | | 0.500 | | | |
| –3 | 1 | | | | | |
| 4 | 1.1300 | 0.7923–1.6117 | | | | |
| Clinical N status | | | 0.966 | | | |
| Absence | 1 | | | | | |
| Presence | 1.0064 | 0.7480–1.3542 | | | | |
| Distant metastasis | | | 0.247 | | | |
| Absence | 1 | | | | | |
| Presence | 1.2784 | 0.8434–1.9378 | | | | |
| Approach | | | 0.427 | | | |
| Laparo | 1 | | | | | |
| Open | 1.1755 | 0.7891–1.7510 | | | | |
| Combined resection | | | 0.938 | | | |
| None | 1 | | | | | |
| Yes | 1.0222 | 0.5859–1.7833 | | | | |
| Operation time, min | | | 0.070 | | | 0.0935 |
| <222 | 1 | | | 1 | | |
| ≥222 | 1.3107 | 0.9785–1.7557 | | 1.2856 | 0.9585–1.7244 | |

*HD*, hemodialysis; *ASA*, American Society of Anesthesiologists; *CD*, Clavien–Dindo.

## Discussion

This multicenter, retrospective study investigated the short-term outcomes of consecutive colon cancer patients who underwent curative resection. The postoperative complication rate was significantly higher in HD than non-HD patients (36.8% vs. 15.5%, p = 0.009). All

postoperative complications in the HD patients were infectious type. Univariate and multivariate analyses revealed HD as a risk factor for postoperative complications.

In a retrospective and multi-institutional study regarding the relationship between dialysis and postoperative outcomes in colorectal cancer, Hu et al. reported that 0.6% of patients were undergoing dialysis [12]. Gajdos et al. performed a retrospective cohort study of the complications and short-term outcomes of elective general surgery in dialysis patients, and found that 0.9% of all patients were dialysis patients and 97.7% of dialysis patients were ASA-PS>3 [13]. In the present study, the proportion of HD patients in the overall cohort was 1.4%, and these patients had poorer PS (ASA-PS>3, n = 19 [100%]) and a higher rate of comorbidities (94.7%).

Dialysis patients undergoing surgery are known to have a high risk of postoperative complications, including infectious complications, acute coronary syndrome (ACS), ischemic bowel disease, and bleeding [10,14]. A previous study reported a complication rate of 41.4% in HD patients after elective abdominal surgery [8], whereas in the present study, postoperative complications occurred in 36.8% of HD patients. Surprisingly, all of the present complications were infectious. Several other studies have also reported a tendency for dialysis patients to develop postoperative infections, especially pneumonia and SSI [12,13]. A possible reason for this tendency is that patients with uremia are more susceptible to infectious agents; in addition, renal anemia and hypoproteinemia suppress the healing process and cause proliferation of fibroblasts and delayed wound healing, resulting in SSI or anastomotic leakage [9,15–17]. Another possible explanation is that dialysis treatment induces diffusion of carbon dioxide in the dialysate, resulting in hypoxemia and reflex hypoventilation, which increases the risk of postoperative atelectasis and pneumonia [18]. The low head position that is often used in colorectal cancer surgery can also cause respiratory complications, including atelectasis; accordingly, anesthesiologists set high positive end-expiratory pressure or allow high airway pressure.

A previous study reported that 33% of dialysis patients and 42% of non-dialysis patients underwent laparoscopic surgery for colorectal cancer surgery, and concluded that in dialysis patients, the laparoscopic approach was associated with fewer postoperative complications, lower mortality, and shorter total length of hospital stay [12]. In the present study, a high proportion of HD patients (78.9%) underwent laparoscopic surgery; compared with open surgery, the laparoscopic approach has a lower risk of infectious complications because it is less invasive and the length of the wound is shorter. However, the rate of infectious complications in the present study was still high, which may be due in part to the long operation time. Accordingly, we consider that surgery for HD patients should be performed in the shortest possible time by an experienced surgical team. In addition to consideration of the best surgical approach, it is also important to pay attention to the perioperative care of HD patients who undergo surgery for colon cancer.

It is noteworthy that the complications of ACS or ischemic bowel disease were not observed in our study. Dialysis patients are likely to develop conditions that promote arteriosclerosis, such as storage of uremic toxins, abnormal metabolism of minerals such as phosphorus and calcium, and increased fluid volume. Accordingly, they often develop vascular complications such as ACS and ischemic bowel disease. In fact, a dialysis regimen has been reported that employs strict fluid management for patients undergoing elective colorectal surgery, with the aim of reducing the rates of postoperative cardiopulmonary and tissue-healing complications [19,20]. Currently, dialysis treatment is often performed the day before and after elective surgery, and advances in intraoperative and postoperative management include such as fluid restriction and the use of vasodilators. In the present study, surgeries for HD patients were performed at high-volume centers that have nephrology and cardiology facilities. It is desirable that surgeries for HD patients should be performed at such well-managed facilities.

This study has several limitations. First, it was a retrospective study and the sample size of HD patients was small. The proportion of HD patients was reported to be ~0.3% in the general population and 0.6% in a colorectal cancer cohort [12]. In the present study, 1.4% of all participants were HD patients, which is similar to previous reports [12,13]. To further examine the number of HD patients, a larger scale study is needed. Second, we analyzed only patients who underwent colon cancer surgery, and excluded those who were unable to undergo surgery due to poor general condition or underwent emergency surgery. Anastomotic leakage is a serious complication and the rate was high in our study. Although recent studies have proposed that blood flow evaluation by indocyanine green is useful for detecting anastomotic leakage, we did not perform this evaluation due to a lack of specific equipment and because performing the test would require the consensus of all attending physicians [21,22]. Further prospective evaluation is needed to resolve this issue. Third, although Hb and albumin are also very important factors, it is difficult to collect additional blood test data because the present data were collected from multiple facilities. However, we calculated variance inflation factors (VIFs) to detect multi-collinearity among the predictors in our regression model. The VIFs were 1.902 (sex), 2.355 (age), 1.279 (BMI), 3.214 (ASA-PS), 1.039 (HD), 1.678 (tumor location), 1.168 (preoperative chemotherapy), 1.832 (clinical T), 2.131 (clinical N), 1.375 (clinical M), 1.345 (combined resection), 1.272 (surgical approach), and 2.132 (operation time). Therefore, we consider that multi-collinearity is not a significant problem in our model.

## Conclusion

It is expected that the number of colon cancer surgeries performed in HD patients will continue to increase. As advances in perioperative management and minimally invasive surgery have become mainstream, it is necessary to remain aware that short-term complications can occur, especially infectious complications in HD patients.

## Supporting information

**S1 File.**
(XLSX)

## Author Contributions

**Data curation:** Shintaro Hashimoto, Kiyoaki Hamada, Masato Araki, Yorihisa Sumida, Hiroaki Takeshita, Hidetoshi Fukuoka, Hideo Wada, Kazuo To, Mariko Yamashita, Kenji Tanaka.

**Supervision:** Tetsuro Tominaga, Takashi Nonaka, Terumitsu Sawai, Takeshi Nagayasu.

**Writing – original draft:** Toshio Shiraishi.

**Writing – review & editing:** Toshio Shiraishi, Tetsuro Tominaga.

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
