## [Decision Letter · Decision Letter 0]

28 Apr 2021

PONE-D-21-10207

Effect of hemodialysis on short-term outcomes after colon cancer surgery

PLOS ONE

Dear Dr. Tominaga,

Thank you for submitting your manuscript to PLOS ONE. After careful consideration, we feel that it has merit but does not fully meet PLOS ONE’s publication criteria as it currently stands. Therefore, we invite you to submit a revised version of the manuscript that addresses the points raised during the review process.

Please revise accordingly.

We look forward to receiving your revised manuscript.

Kind regards,

Academic Editor

PLOS ONE

Journal Requirements:

"Nagasaki University Hospital Clinical Research Ethics Committee

Permission number : 16062715-2

We gave the patients written and oral explanation using the consent statement and obtained consent signature document."

a) Please amend your current ethics statement to confirm that your named institutional review board or ethics committee specifically approved this study.

Reviewers' comments:

Reviewer's Responses to Questions

**Comments to the Author**

1. Is the manuscript technically sound, and do the data support the conclusions?

Reviewer #1: Yes

Reviewer #2: Partly

Reviewer #3: Partly

2. Has the statistical analysis been performed appropriately and rigorously? 

Reviewer #1: I Don't Know

Reviewer #2: Yes

Reviewer #3: No

3. Have the authors made all data underlying the findings in their manuscript fully available?

Reviewer #1: No

Reviewer #2: Yes

Reviewer #3: Yes

4. Is the manuscript presented in an intelligible fashion and written in standard English?

Reviewer #1: Yes

Reviewer #2: No

Reviewer #3: No

5. Review Comments to the Author

Reviewer #1: Since “Hemodialysis patients who undergo surgery have a high risk of postoperative complications” has well documented in many studies, conclusions of this study did not provide new information for readers. I teject this study due to lack of novelty.

Reviewer #2: In this manuscript, Shiraishi and colleagues used a multicenter and retrospective analysis to examine the association between hemodialysis patients and elective curative resection surgery for colon cancer based on the multicenter and retrospective study. This research involved 1165 patients who had colon cancer surgery and included 19 patients on hemodialysis, 59 patients with renal disfunction, and 1087 patients who had normal renal function. Male sex and hemodialysis had a significant correlation in postoperative complications in patients following colon cancer surgery, according to the univariate and multivariate analysis. The following are my questions and recommendations for the authors to consider improving the manuscript:

1. The aim of this study is to explore whether colon cancer surgery can be done safely in hemodialysis patients. The hemodialysis patients are the main topic, but the samples size of hemodialysis analyzed is slightly smaller. While I am aware that the authors include a summary in the discussion section, may key question remains.

Please provide a more detailed explanation about the blind spots caused by this point.

2. In Table 4, the univariate and multivariate analysis revealed sex had a significantly association with complications. Please clarify this in the discussion. In addition, Table 4 is difficult to understand. For example, authors mentioned that univariate analysis revealed male sex (Odds ratios, 1.8537; 95%CI, 1.3445-2.5556; p<0.001) and HD (Odds ratio, 2.9593; 95%CI, 1.1163-7.8451; P=0.029) are risk factors significantly associated with complications in the result section. Please confirm in Table 4.

3. In the result section, authors mentioned that the rate of patients with postoperative complications (CD>2) was 36.5% (n=7) in the hemodialysis group, 18.6% (n=11) in the renal disfunction group, and 16.2% (n=176) in the normal group. This result is from Table 3, not in Table 2. Please confirm if the rate of patients with postoperative complications in the hemodialysis group is 36.5% or 36.8%.

5. Please revise the abstract to explain the research’s findings and conclusions more clearly. The abstract need to be remodeled.

6. The manuscript is not well written and requires an English language editing

Reviewer #3: # The major concern is the definition of the outcome. How were the outcomes measured? In the Methods session, the authors defined the postoperative complications as patients with Clavien–Dindo (CD) grade 2 or higher that occurred within 30 days of the primary surgery. However, the author did not indicate the definitions of the complications clearly and what is Clavien–Dindo (CD) grade 2 or higher. In the Results session (Table 3), the authors listed the complications they included, but they still not defined them clearly. In addition, is anastomotic leak usually defined as infectious complication as shown in the Table 3, or is it related to technique problem and leading to secondary infection? Furthermore, one of the post-op complications among HD patients is UTI. The concern is that is this patient truly has UTI or just asymptomatic pyuria?

# In the Methods session, the authors excluded patients with incomplete laboratory data or synchronous colon cancer; 1165 patients were included in the final analysis. However, the authors did not show that how many patients were evaluated initially and how many were excluded due to incomplete laboratory data and synchronous colon cancer, respectively? Are there any differences in characteristics between those who were excluded and those who included in the analysis? In addition, in the results session, the authors did not present any lab data. Why did they need to exclude patients with incomplete laboratory data? I think it is more reasonable that the authors included those patients in the analysis and treated incomplete laboratory data as missing data.

# In the Methods session, the authors should define renal disfunction more clearly. What are the definitions of proteinuria and low glomerular filtration rate according to blood and urinary tests? In addition, the authors did not indicate clearly what comorbidities did they include?

# In the statistical analysis of Methods session, there is a sentence: “Differences in continuous variables were analyzed with the chi-squared test.” This is a wrong statement.

# In the Table 4, the author did not show the odds ratio and 95% C.I. of the univariable analysis. It is kind of weird. In addition, the authors did not indicate what variables were adjusted in the multivariable logistic regression. Particularly, comorbidity such as diabetes mellitus may be confounders.

# The authors concluded that to reduce the risk of postoperative complications, it is important that surgery for HD patients is performed in facilities with experienced surgical teams and excellent perioperative management. The authors did not evaluate whether experienced surgical teams and perioperative management were associated with less complications among HD patients, so they should not draw this conclusion.

6. PLOS authors have the option to publish the peer review history of their article (what does this mean?). If published, this will include your full peer review and any attached files.

Reviewer #1: No

Reviewer #2: No

Reviewer #3: Yes; Dr. Ming-Yan Jiang

---

## [Author Response · Author response to Decision Letter 0]

28 Sep 2021

Response to Reviewer #1: 

Since “Hemodialysis patients who undergo surgery have a high risk of postoperative complications” has well documented in many studies, conclusions of this study did not provide new information for readers. I teject this study due to lack of novelty.

Response: As you pointed out, previous studies have reported the problem of increased risk of short-term complications in HD patients. However, in recent years the spread of minimally invasive surgery and improvements in perioperative management have contributed to a reduction in the complication rate. In the present manuscript, we report the current status of surgery for HD patients, with relatively recent data.

Response to Reviewer #2: 

1. The aim of this study is to explore whether colon cancer surgery can be done safely in hemodialysis patients. The hemodialysis patients are the main topic, but the samples size of hemodialysis analyzed is slightly smaller. While I am aware that the authors include a summary in the discussion section, may key question remains.

Please provide a more detailed explanation about the blind spots caused by this point.

Response: The proportion of HD patients was reported as ~0.3% in the general population, and 0.6% in a colorectal cancer cohort (Hu WH et al., Int J Colorectal Dis 2015;30:1557-1562). In the present study, 1.4% of all participants were HD patients, which is similar to previous reports (Hu WH et al., Int J Colorectal Dis 2015;30:1557-1562) (Gajdos C et al., JAMA Surg. 2013; 148(2): 137-143). To further examine the number of HD patients, a larger scale study is needed. We have added these sentences to the Limitations section, and rewrote the sentence “Postoperative complications were most frequent in HD patients, and were infectious in all cases.” to “The postoperative complication rate was significantly higher in HD than non-HD patients (36.8% vs. 15.5%, p = 0.009). All postoperative complications in the HD patients were infectious type.” in the Discussion section.

2. In Table 4, the univariate and multivariate analysis revealed sex had a significantly association with complications. Please clarify this in the discussion. In addition, Table 4 is difficult to understand. For example, authors mentioned that univariate analysis revealed male sex (Odds ratios, 1.8537; 95%CI, 1.3445-2.5556; p<0.001) and HD (Odds ratio, 2.9593; 95%CI, 1.1163-7.8451; P=0.029) are risk factors significantly associated with complications in the result section. Please confirm in Table 4.

Response： I apologize for this error. We have corrected “univariate” to “multivariate” in this sentence. As pointed out by reviewer #3, we did not show how many patients were evaluated initially and how many were excluded due to incomplete laboratory data and synchronous colon cancer, respectively. Accordingly, we performed the statistical analysis again in the total of 1372 cases, including those with missing data. Although there was almost no change in the content, “male sex” was not used in multivariate analysis because there was no significant difference in the univariate analysis (p > 0.20). Therefore, we deleted the reference to “male sex” in Table 4 from the Results and Discussion sections.

3. In the result section, authors mentioned that the rate of patients with postoperative complications (CD>2) was 36.5% (n=7) in the hemodialysis group, 18.6% (n=11) in the renal disfunction group, and 16.2% (n=176) in the normal group. This result is from Table 3, not in Table 2. Please confirm if the rate of patients with postoperative complications in the hemodialysis group is 36.5% or 36.8%.

Response： We apologize for this error. The complication rate in Table 2 was incorrect, and that in Table 3 was correct. We have corrected “36.5%” to “36.8%” in this sentence.

5. Please revise the abstract to explain the research’s findings and conclusions more clearly. The abstract need to be remodeled.

Response： Thank you this suggestion. We have modified the abstract and clarified the conclusion accordingly.

6. The manuscript is not well written and requires an English language editing

Response: The manuscript has been edited by a professional native-English-speaking editor (FORTE Science Communications, https://www.fortescience.com/) and a certificate of editing is provided.

Response to Reviewer #3: 

# The major concern is the definition of the outcome. How were the outcomes measured? In the Methods session, the authors defined the postoperative complications as patients with Clavien–Dindo (CD) grade 2 or higher that occurred within 30 days of the primary surgery. However, the author did not indicate the definitions of the complications clearly and what is Clavien–Dindo (CD) grade 2 or higher. In the Results session (Table 3), the authors listed the complications they included, but they still not defined them clearly. In addition, is anastomotic leak usually defined as infectious complication as shown in the Table 3, or is it related to technique problem and leading to secondary infection? Furthermore, one of the post-op complications among HD patients is UTI. The concern is that is this patient truly has UTI or just asymptomatic pyuria?

Response: Thank you for this important comment. As suggested, we have added definitions for each complication and condition of CD>2 in the Materials and Methods section. We classified anastomotic leak as an infectious complication because it occurs despite measures against suture failure such as leak test, reinforcement, and blood flow confirmation with ICG at the anastomotic site. We included one case of urinary tract infection in the postoperative complications of the HD group, that was diagnosed based on the laboratory findings in a patient with residual urine.

# In the Methods session, the authors excluded patients with incomplete laboratory data or synchronous colon cancer; 1165 patients were included in the final analysis. However, the authors did not show that how many patients were evaluated initially and how many were excluded due to incomplete laboratory data and synchronous colon cancer, respectively? Are there any differences in characteristics between those who were excluded and those who included in the analysis? In addition, in the results session, the authors did not present any lab data. Why did they need to exclude patients with incomplete laboratory data? I think it is more reasonable that the authors included those patients in the analysis and treated incomplete laboratory data as missing data.

Response: I apologize for the incorrect sentence “incomplete laboratory data” in the Materials and Methods, which should have been “incomplete clinicopathological data”. In response to the reviewer’s suggestion, we performed the statistical analysis again, in a total of 1372 cases, including those with missing data. We excluded 15 patients with synchronous colon cancer from the total 1387 patients whose data were collected. We have checked all Tables accordingly, but made almost no change to the content. Although multivariate analysis revealed “male sex” as a risk factor significantly associated with complications at the time of the first submission, “male sex” was not used in the multivariate analysis this time because there was no significant difference in the univariate analysis and p >0.20, unlike last time. Multivariate analysis revealed HD (Odds ratio, 2.9362; 95%CI, 1.1384–7.5730; p=0.026) as a risk factor significantly associated with complications, as in the manuscript submitted initially.

# In the Methods session, the authors should define renal disfunction more clearly. What are the definitions of proteinuria and low glomerular filtration rate according to blood and urinary tests? In addition, the authors did not indicate clearly what comorbidities did they include?

Response: We have added definitions for renal dysfunction, urinary infection, and condition of CD>2 in the Materials and Methods section, as follows:

“In Japan, chronic kidney disease is generally defined as persistence for �3 months of one or more of the following: obvious renal dysfunction in pathological examination, estimated glomerular filtration rate <60 ml/min/1.73m2 in blood test, or proteinuria in urinalysis. In this study, renal dysfunction was defined as chronic kidney disease without HD.” 

“Comorbidities included such as hypertension, diabetes, heart disease, and cerebrovascular disease.”

“Urinary tract infection was defined as pyuria according to urinalysis. Among the postoperative complications in the HD group, urinary tract infection was defined as patients with urine output who had findings of urinary tract infection.”

# In the statistical analysis of Methods session, there is a sentence: “Differences in continuous variables were analyzed with the chi-squared test.” This is a wrong statement.

Response: I apologize for this incorrect sentence, which has been deleted.

# In the Table 4, the author did not show the odds ratio and 95% C.I. of the univariable analysis. It is kind of weird. In addition, the authors did not indicate what variables were adjusted in the multivariable logistic regression. Particularly, comorbidity such as diabetes mellitus may be confounders.

Response: In response to the reviewer’s suggestion, we have added the odds ratio and 95% C.I. of the univariable analysis to Table 4, and added the sentence “Clinical variables with a p value <0.2 in the univariate analysis were included in the multivariate analysis.” to the Materials and Methods section. We appreciate the reviewer’s important comment about confounders. As pointed out, HD was significantly associated with diabetes mellitus (p = 0.011; Fisher’s exact test). 

We calculated variance inflation factors (VIFs) to detect multi-collinearity among the predictors in our regression models. The VIFs were 1.902 (sex), 2.355 (age), 1.279 (BMI), 3.214 (ASA-PS), 1.039 (HD), 1.678 (tumor location), 1.168 (preoperative chemotherapy), 1.832 (clinical T), 2.131 (clinical N), 1.375 (clinical M), 1.345 (combined resection), 1.272 (surgical approach), and 2.132 (operation time). Therefore, we consider that multi-collinearity is not a significant problem in our model.

# The authors concluded that to reduce the risk of postoperative complications, it is important that surgery for HD patients is performed in facilities with experienced surgical teams and excellent perioperative management. The authors did not evaluate whether experienced surgical teams and perioperative management were associated with less complications among HD patients, so they should not draw this conclusion.

Response: Thank you for this important comment. We have modified the Abstract and clarified the Conclusion accordingly. We have added the following sentence to the Abstract: “Despite recent advances in perioperative management and minimally invasive surgery, it is necessary to be aware that short-term complications can still occur, especially infectious complications in hemodialysis patients.”

---

## [Decision Letter · Decision Letter 1]

1 Nov 2021

PONE-D-21-10207R1Effect of hemodialysis on short-term outcomes after colon cancer surgeryPLOS ONE

Dear Dr. Tominaga,

Thank you for submitting your manuscript to PLOS ONE. After careful consideration, we feel that it has merit but does not fully meet PLOS ONE’s publication criteria as it currently stands. Therefore, we invite you to submit a revised version of the manuscript that addresses the points raised during the review process.

Please revise.

We look forward to receiving your revised manuscript.

Kind regards,

Academic Editor

PLOS ONE

Reviewers' comments:

Reviewer's Responses to Questions

**Comments to the Author**

1. If the authors have adequately addressed your comments raised in a previous round of review and you feel that this manuscript is now acceptable for publication, you may indicate that here to bypass the “Comments to the Author” section, enter your conflict of interest statement in the “Confidential to Editor” section, and submit your "Accept" recommendation.

Reviewer #2: All comments have been addressed

Reviewer #3: (No Response)

2. Is the manuscript technically sound, and do the data support the conclusions?

Reviewer #2: Yes

Reviewer #3: No

3. Has the statistical analysis been performed appropriately and rigorously? 

Reviewer #2: Yes

Reviewer #3: No

4. Have the authors made all data underlying the findings in their manuscript fully available?

Reviewer #2: Yes

Reviewer #3: No

5. Is the manuscript presented in an intelligible fashion and written in standard English?

Reviewer #2: Yes

Reviewer #3: No

6. Review Comments to the Author

Reviewer #2: (No Response)

Reviewer #3: Major

#1. Why did the authors include those complications? What are they based on?

#2. The authors mentions that the variables with a p value < 0.2 in the univariate analysis were included in the multivariate analysis. That is, in the regression analysis in table 4, they only included HD and operation time in the multivariate model. As shown in their table 2, HD group is associated with comorbidities, which could be confounders and should be adjusted in the regression model. In addition, anemia and malnutrition were also risk factors of post-operational complications. I suggest the authors also to collect lab data such as Hb and albumin and adjust these potential confounders.

Minor

#1. The authors should mention what comorbidity they included in the method session or table 1 and table 2.

#2. In table 4, the authors divided continuous variables such as age, BMI, and operation time into binary variables. They also divided the ordinal variable ASA performance status into binary variable. What do they based on to decide the cut points?

#3. In the methods session, the author indicated the definitions of CKD. What did they mean by “obvious renal dysfunction in pathological examination”? How are the estimated glomerular filtration rates calculated?

7. PLOS authors have the option to publish the peer review history of their article (what does this mean?). If published, this will include your full peer review and any attached files.

Reviewer #2: No

Reviewer #3: **Yes: **Ming-Yan Jiang

---

## [Author Response · Author response to Decision Letter 1]

30 Nov 2021

Response to reviewers’ comments

November 30, 2021

Academic Editor

PLOS ONE

Dear Dr. Chen,

Thank you for considering our paper entitled “Effect of hemodialysis on short-term outcomes after colon cancer surgery” (PONE-D-21-10207) for publication in PLOS ONE, subject to revision. We have revised our manuscript according to the helpful comments provided by the reviewers, addressed the concerns raised, and accepted their valuable advice.

Changes in the revised manuscript are shown in bold red font. Our point-by-point responses to the reviewers’ comments are provided below.

Response to Reviewer 3

Major

#1. Why did the authors include those complications? What are they based on?

Response:

We chose the postoperative complications listed in the Clavien Dindo (CD) classification. This time, we focused on CD grades 2 and higher because these are clinically problematic. We have added the following sentence to the Materials and methods section: “This time, we focused on CD grades 2 and higher because these are clinically problematic.”

#2. The authors mentions that the variables with a p value < 0.2 in the univariate analysis were included in the multivariate analysis. That is, in the regression analysis in table 4, they only included HD and operation time in the multivariate model. As shown in their table 2, HD group is associated with comorbidities, which could be confounders and should be adjusted in the regression model. In addition, anemia and malnutrition were also risk factors of post-operational complications. I suggest the authors also to collect lab data such as Hb and albumin and adjust these potential confounders.

Response:

Thank you for this important comment. We calculated variance inflation factors (VIFs) to detect multi-collinearity among the predictors in our regression models. The VIFs were 1.902 (sex), 2.355 (age), 1.279 (BMI), 3.214 (ASA-PS), 1.039 (HD), 1.678 (tumor location), 1.168 (preoperative chemotherapy), 1.832 (clinical T), 2.131 (clinical N), 1.375 (clinical M), 1.345 (combined resection), 1.272 (surgical approach), and 2.132 (operation time). Therefore, we consider that multi-collinearity is not a significant problem in our model. We agree with the reviewer that Hb and albumin are very important factors. However, it is difficult to collect additional blood test data because we have dealt with data from multiple facilities this time. Therefore, we will mention this point as a limitation.

In response, we have added the following to the limitation section: “Although Hb and albumin are also very important factors, it is difficult to collect additional blood test data because the present data were collected from multiple facilities. However, we calculated variance inflation factors (VIFs) to detect multi-collinearity among the predictors in our regression model. The VIFs were 1.902 (sex), 2.355 (age), 1.279 (BMI), 3.214 (ASA-PS), 1.039 (HD), 1.678 (tumor location), 1.168 (preoperative chemotherapy), 1.832 (clinical T), 2.131 (clinical N), 1.375 (clinical M), 1.345 (combined resection), 1.272 (surgical approach), and 2.132 (operation time). Therefore, we consider that multi-collinearity is not a significant problem in our model.”

Minor

#1. The authors should mention what comorbidity they included in the method session or table 1 and table 2.

Response:

Table 1 lists all comorbidities that were assessed, including hypertension, diabetes, dyslipidemia, heart disease, and cerebrovascular disease. Table 2 shows the presence or absence of the same comorbidities as in Table 1, except for HD and renal dysfunction. We have clarified this point in the Materials and methods section.

#2. In table 4, the authors divided continuous variables such as age, BMI, and operation time into binary variables. They also divided the ordinal variable ASA performance status into binary variable. What do they based on to decide the cut points?

Response:

In Japan, ‘elderly’ is defined as age 65 and over, but many people have good PS even if they are aging and have comorbidities. Therefore, it is common in Japan that those aged 70 and over are considered ‘elderly’ when comparing the results statistically.

The BMI considered to indicate obesity differs among countries. In Japan, BMI �25 is generally used as the standard, and for this reason we set the BMI cutoff point to 25. 

Regarding operation time, we decided the cutoff point using the ROC curve tool of Bell Curve for Excel software.

We divided ASA-PS into 1 and 2 or more to enable comparison of patients who had good general condition without comorbidities with those who had some comorbidities.

#3. In the methods session, the author indicated the definitions of CKD. What did they mean by “obvious renal dysfunction in pathological examination”? How are the estimated glomerular filtration rates calculated?

Response:

Thank you for this important comment. Many kidney diseases are confirmed by histopathology, and the histopathological findings lead to estimation of renal function prognosis and treatment policy decisions. I apologize that this sentence was incorrect. We have replaced “obvious renal dysfunction” with “obvious kidney disease”. In Japan, the following formula is generally used to calculate eGFR:

eGFR = 194 × serum Cr-1.094 × age-0.287 (× 0.739 for females).

Please also note that we have identified an error in Table 1. Although not directly related to the main focus of this article, the number has been corrected. We apologize for this oversight.

I confirm that all authors have approved the revised manuscript. We hope that we have satisfied the concerns of the reviewers, and that the manuscript will now be considered suitable for publication in PLOS ONE. Please feel free to contact me if any questions arise.

Sincerely,

Tetsuro Tominaga, MD, PhD

Department of Surgical Oncology

Nagasaki University Graduate School of Biological Sciences

1-7-1 Sakamoto, Nagasaki 852-8501, Japan

Phone: +81-95-819-7304 Fax: +81-95-819-7306

E-mail: tetsuro.tominaga@nagasaki-u.ac.jp

---

## [Decision Letter · Decision Letter 2]

28 Dec 2021

Effect of hemodialysis on short-term outcomes after colon cancer surgery

PONE-D-21-10207R2

Dear Dr. Tominaga,

We’re pleased to inform you that your manuscript has been judged scientifically suitable for publication and will be formally accepted for publication once it meets all outstanding technical requirements.

Kind regards,

Academic Editor

PLOS ONE

Additional Editor Comments (optional):

Reviewers' comments:

Reviewer's Responses to Questions

**Comments to the Author**

1. If the authors have adequately addressed your comments raised in a previous round of review and you feel that this manuscript is now acceptable for publication, you may indicate that here to bypass the “Comments to the Author” section, enter your conflict of interest statement in the “Confidential to Editor” section, and submit your "Accept" recommendation.

Reviewer #2: All comments have been addressed

Reviewer #4: All comments have been addressed

2. Is the manuscript technically sound, and do the data support the conclusions?

Reviewer #2: Yes

Reviewer #4: Yes

3. Has the statistical analysis been performed appropriately and rigorously? 

Reviewer #2: Yes

Reviewer #4: Yes

4. Have the authors made all data underlying the findings in their manuscript fully available?

Reviewer #2: Yes

Reviewer #4: Yes

5. Is the manuscript presented in an intelligible fashion and written in standard English?

Reviewer #2: Yes

Reviewer #4: Yes

6. Review Comments to the Author

Reviewer #2: (No Response)

Reviewer #4: (No Response)

7. PLOS authors have the option to publish the peer review history of their article (what does this mean?). If published, this will include your full peer review and any attached files.

Reviewer #2: No

Reviewer #4: No

---

## [Editor Report · Acceptance letter]

3 Jan 2022

PONE-D-21-10207R2 

Effect of hemodialysis on short-term outcomes after colon cancer surgery 

Dear Dr. Tominaga:

I'm pleased to inform you that your manuscript has been deemed suitable for publication in PLOS ONE. Congratulations! Your manuscript is now with our production department. 

Kind regards, 

on behalf of

Dr. Robert Jeenchen Chen 

Academic Editor

PLOS ONE